# Poor Cognitive Agility Conservation in Obese Aging People

**DOI:** 10.3390/biomedicines11010138

**Published:** 2023-01-05

**Authors:** Teresa Pardo-Moreno, Himan Mohamed-Mohamed, Antonio Rivas-Dominguez, Victoria Garcia-Morales, Ruben A. Garcia-Lara, Sami Suleiman-Martos, Beatriz Bermudez-Pulgarin, Juan Jose Ramos-Rodriguez

**Affiliations:** 1Department of Physiology, Faculty of Health Sciences (Ceuta), University of Granada, 51001 Ceuta, Spain; 2Instituto Nacional de Gestión Sanitaria (INGESA), Primary Health Care, 51003 Ceuta, Spain; 3Department of Celular Biology, University of Seville, 41009 Seville, Spain; 4Department of Biomedicine, Biotechnology and Public Health, Physiology Area, Faculty of Medicine, University of Cádiz, Pl. Falla, 9, 11003 Cádiz, Spain; 5UGC Orgiva, Granada-South Helth Management Area, Andalusian Health Service, 18420 Granada, Spain; 6Servicio Andaluz de Salud, Av. Del Sur 11, 18014 Granada, Spain

**Keywords:** obesity, metabolic syndrome, cognitive function, dementia, mental agility

## Abstract

Life expectancy has been boosted in recent decades at expenses of increasing the age-associated diseases. Dementia, for its incidence, stands out among the pathologies associated with aging. The exacerbated cognitive deterioration disables people from carrying out their daily lives autonomously and this incidence increases exponentially after 65 years of age. The etiology of dementia is a miscellaneous combination of risk factors that restrain the quality of life of our elderly. In this sense, it has been established that some metabolic pathologies such as obesity and diabetes act as a risk factor for dementia development. In contrast, a high educational level, as well as moderate physical activity, have been shown to be protective factors against cognitive impairment and the development of dementia. In the present study, we have evaluated the metabolic composition of a population between 60–90 years old, mentally healthy and with high academic degrees. After assessing agility in mental state, we have established relationships between their cognitive abilities and their body composition. Our data support that excess body fat is associated with poorer maintenance of cognition, while higher percentages of muscle mass are associated with the best results in the cognitive tests.

## 1. Introduction

Obesity is considered one of the biggest pandemics currently plaguing industrialized countries [1]. The elderly is the population hit the hardest by this sickness [2]. Due to this upward trend, by 2025 it is estimated that the ratio of men and women with obesity will exceed 18% and 21%, respectively [3]. In addition, it has been observed that annual deaths due to obesity and overweight reached 28 million victims worldwide [4]. This growing prevalence has a significant impact on national health systems. Medical care cost in patients with obesity represents 30% more per year than patients with a normal Body Mass Index (BMI) [1]. In line with this, it has also been shown that a 1% increase in BMI, at the age of 70, could increase the risk of Alzheimer’s disease (AD) by 36% [5]. Therefore, the risk of developing cognitive impairment due to obesity is directly associated in people older than 65 years [6]. In this sense, hypercaloric diets have been related with insulin resistance, prediabetes, and Type 2 Diabetes (T2D).

Chronic inflammatory diseases such as hypertension, hypercholesterolemia, and insulin resistance, which is the prelude to T2D, are strongly related to obesity development and contribute to the appearance of the well-known metabolic syndrome [7,8]. In this context, insulin receptors are widely distributed throughout the central nervous system [9]. They are localized in astrocytes, endothelial cells, and neuronal synapses. Moreover, they are very abundant in the hippocampus, cortex, and cerebellum, protecting neurons from neurodegeneration and cell death, thus memory and learning processes. Thereby, insulin resistance associated with obesity may cause an imbalance in neural metabolism (for review [10]).

From another point of view, insulin resistance detected in obese patients could be mediated by a decrease in the activity of the enzymes responsible for its degradation, such as insulin degrading enzyme and neprylisin. Both enzymes are involved not only in insulin degradation, but also in amyloid-β degradation in the central nervous system. Amyloid-β accumulation is the first milestone in AD development. In this sense, a possible mismatch in the insulin degradation by those enzymes may influence the pathogenesis of AD, promoting an increase of amyloid-β accumulation and neuronal death [11].

On the other hand, obesity and its comorbidities are also risk factors associated with impaired cognitive performance and cognitive decline. The onset of cognitive decline is evident at young age of 45 years with a 3.6 % decline in mental reasoning demonstrated in both male and female populations [7,12,13,14]. A negative association has been shown between anthropometric measures like BMI and cognitive function. For instance, obesity decreases performance on tasks of episodic memory. Verbal learning, indexed by delayed recall and recognition of words, worsen with increasing BMI [15,16]. Furthermore, it seems that some studies showed that obese young adults have impaired working memory [17]. Some studies find no difference in memory performance between obese and non-obese individuals [18], and even some of these studies indicate that obesity even seems to offer a level of protection against cognitive decline [19,20]. Contrarily, some others suggest that there is a strong relationship between obesity and cognitive function with a gender-dependent behavioral difference [21]. The relationship between obesity and later cognitive decline is not totally clear yet. Despite the studies carried out to date that reflect a detriment on cognitive performance in obese adults (aged 18–65 years) [22,23], there is insufficient evidence to support that fact. In this sense, the FINGER-study was the first to show that a multidomain lifestyle intervention is beneficial for prevention of cognitive decline [24]. Since 2009, when this project began, the strategies based on changes in multidomain lifestyle intervention prevent cognitive decline have been refuted [25,26]. 

The Mini–Mental State Examination (MMSE) is a simple and widely accepted test that allows to observe the presence of cognitive impairment associated with dementia, for example, incipient AD. This test is very useful for detecting mild cognitive impairment and/or possible dementias in geriatric patients. In addition, it corrects the score according to the age of each patient, thus differentiating age-specific mental deterioration or slowing associated with dementia.

Available studies that correlate cognitive impairment with body composition in mentally active older people are currently very scarce. In this sense, the aim of this study was to explore whether age-related cognition abilities, evaluated with the MMSE, could be affected by obesity in elderly (60–90 years old) without a clinical diagnosis of dementia. We differentiated three working groups based on the results obtained with the mini-mental test: (1) scores greater than 95% of the total score, (2) scores between 95–90%, and (3) scores below 90% (without admitting values equivalent to a dementia). Our results have shown that the group with better performance in the cognitive test, considering the anthropometric values (BMI, % body fat, % visceral fat, and % muscle mass), coincides with those subjects with a higher muscle mass volume and a lower body fat index, while the poor cognition scores were associated with obesity anthropometric profile. Interestingly, metabolic parameters as well as BMI or percentage of body fat resulted in reliable predictors in cognition scores. Moreover, the muscle mass loss also seems to be associated with poor punctuation in the MMSE.

## 2. Materials and Methods

### 2.1. Study Design and Population

Our study was carried out in a population aged between 60–90 years. All the individuals belonged to the permanent elderly class and/or were employees of the University Campus of Ceuta of the University of Granada. All the participants were mentally active and none of them had any diagnosed psychiatric pathology. Data collection was carried out between January and June 2022. 

### 2.2. Study Groups

To study the relation between cognition preservation and anthropometrics profile, the participants were subdivided into three groups based on the results obtained by the NORMACODERM MMSE version [27], based in the original version described by Folstein et al. [28]. This test was performed to assess the cognitive capacity of the participants by a single nurse trained by a psychologist specializing in dementia. None of them obtained results compatible with those suffering from dementia. The scores obtained were expressed as a percentage, corrected by the maximum grade given to each age group. NORMACORM MMSE version applies a score correction based on the years of academic studies and age (over or under 75 years). This causes that the maximum score to be slightly different in people over 75 years than a younger population. To correct for these slight differences in scores for different ages, we have placed the scores as a reference of 100% of the maximum grade that the patients potentially can obtained. According to the results, we differentiated the following groups:

Group 1: scores higher than 95% of the maximum grade (G+95%) 

Group 2: scores between 95 − 90% of the highest grade (G95 − 90%) 

Group 3: scores less than 90% of the highest grade (G − 90%) 

### 2.3. Anthropometric Measurements and Bioelectrical Impedance Analysis

All the participants were summoned with light clothing to proceed to the study of body composition through the anthropometric measurements of height and weight, through which the BMI was calculated. Subsequently, the percentage of muscle mass, percentage of body fat, percentage of visceral fat, and metabolic consumption were measured by bioelectrical impedance analysis used devices OMRON Healthcare BF 511, as previously described [29,30]. 

### 2.4. Statistical Analysis and Correlation Studies

Metabolic assessment and anthropometric parameters studies were analyzed by one-way ANOVA for independent samples followed by Tuckey b test or Tamhane test as required. Spearman rank’s correlations were used to perform correlation studies MMSE Scores and BMI as well as to explore correlations between percentage of body fat, percentage of visceral fat, and percentage of muscle mass.

### 2.5. Ethical Aspects of Research

The present study has been carried out after obtaining the permission of the Granada Provincial Ethics Committee CEIM/CEI (Junta de Andalucía) with the approval code TFG-EnvRegC-2022. During the study, the confidential treatment of the data was ensured, guaranteeing the anonymity of the participants. All participants were previously informed and voluntarily agreed to participate in the research through informed consent

## 3. Results

As expected, most of the participants recruited had a higher education level. All of them are studying in the permanent elderly class at the University or were employees of the University. A proportion of 50% of participants were female and 50% were male. The distribution by gender and ages in each group can be seen in Table 1. No differences were found in the ages per group.

### 3.1. Body Height, Body Weight and Basal Metabolic Rate

Metabolic parameters including Body Height, Body Weight, and Basal Metabolic Rate were determined in the experiments by bioelectrical impedance analysis. Differences were detected by one-way ANOVA followed by Tuckey-b test. When we compared all study groups, we observed a significant reduction in Body Height in the group with a poor score in the MMSE (G − 90% F (6.53) = 71.64, ** *p* < 0.01 vs. G + 95% and G95 − 90%). Body weight were significantly higher in G95 − 90% when compared with the rest of the groups (F (8.702) = 165.92, ** *p* < 0.01 vs. G + 95% and G − 90%). However, no significant differences were detected in the Basal Metabolic Rate (Table 2). 

### 3.2. Body Composition

Body composition was evaluated by bioelectrical impedance analysis in all the groups subjected to the MMSE Scores. Differences were determined by ANOVA one-way test followed by Tuckey b. BMI was significantly increased in groups with poor MMSE percentage scores. In this sense, G + 95% was the group with the best score, and showed a lower BMI than the rest of the groups [F(11.08) = 15.23, ** *p* < 0.01 vs. G+95%] (Figure 1a). A similar profile was observed with the visceral fat. We found a significant reduction of the scores in the MMSE with higher weight of visceral fat (Figure 1b) [F(7.12) = 22.37, ** *p* < 0.01 vs. G+95%].

Body fat was significantly increased in groups with poor MMSE percentage scores (Figure 2). We found an upward trend of body fat in G95 − 90% and higher increase in G − 90% groups, however significant differences were only detected in the G + 95% compared to the rest of the groups by ANOVA test and Tuckey-b post hoc [F(6.87) = 75.11, ** *p* < 0.01 vs. G+95%]. In contrast, our data also showed a progressive loss of muscle mass in those with the lowest score in the MMSE. We also established a significant reduction in G95 − 90% and G − 90% when compared with G + 95% groups [Tuckey-b: F(8.17) = 20.15, ** *p* < 0.01 vs. G+95%].

### 3.3. Morphological Composition and with Mental Agility Conservation Correlation 

Our previous data (Figure 1 and Figure 2) suggest that morphological body composition parameters may serve as predictors of mental status in the elderly. We found that BMI, percentage of body fat, and percentage of visceral fat were also negatively correlated with cognitive preservation (Figure 3a,b,d), and that the percentage of mass muscle was positively correlated (Figure 3c). Statistical differences were found by Spearman’s rank correlation represented in Figure 3 as follows: (a) BMI/%MMSE score: −0.448 **, (b) % Body Fat/% MMSE score: −0.418 **; (c) Muscle/% MMSE score: 0.414 **, (d) % Visceral Fat /% MMSE: −0.355 **.

All this data suggests that increased body fat mass is correlated with a worse state of cognitive preservation in healthy older people, while expanding muscle mass correlates with a good state of cognitive preservation within the same age group.

## 4. Discussion

The harshness of complications associated with memory loss and the onset of dementia are socially and economically unaffordable. Currently, early diagnosis of the main types of dementia is not being carried out effectively, with the main types of dementia being AD and vascular dementia. Due to the restricted biomarkers available for proper diagnosis, dementias and cognitive impairment are only detectable once the disease has been already established and evolved in the long-term [31]. This situation makes it necessary to explore new prognostic alternatives.

An important fact that we must consider when evaluating our results obtained is their origin and sampling. We are aware that one of the great difficulties found in our study has been the recruitment of a greater number of subjects between the ages of 60 and 90 who were mentally active and without diagnosed cognitive pathologies. The sample reflected here were students enrolled in the permanent classroom for the elderly at the University of Granada on the Ceuta Campus. Even though we found that having a higher education level and mental activity are preventive factors for suffering dementia, they do not avoid its development [32]. Hence, it is still important to study the other factors that may contribute to cognitive decline in people who remain mentally active.

Previous clinical and basic studies have shown that suffering from metabolic disorders such as diabetes and obesity may be involved in the development of dementia [6,33,34,35]. Our findings support the existence of a relationship between anthropometric composition and an excess of body fat with cognitive preservation in cognitively healthy people over 60 years of age.

Aging remains the main risk factor for various types of diseases, including mostly, neurodegenerative diseases. One of the most frequent consequences of aging is the decrease in executive control, attention, learning processes, and memory [36]. This association is based on the physiological changes due to cellular senescence, chronic inflammation, and the imbalance in the cellular redox state [37]. This progressive deterioration can be accelerated with an unhealthy and sedentary lifestyle. Increased physical activity, as well as a healthy diet, may delay the deterioration associated with age [38].

From our side, the individuals with the worst scores in the MMSE showed a lower height than the rest of the groups. Several studies with similar results have proposed that the genetic factors influencing cognitive performance and height may be concurrent, and genetically height could predisposed individuals for better cognitive performance [39]. A study conducted in our neighboring country, Portugal, evidenced that height could be an independent predictor of cognitive function [40]. In this sense, it has been proposed that taller people could be related to greater cognitive reserve, which would delay the clinical expression of dementia [41,42].

As have been described previously, higher BMI predicted declining cognition in individuals over 50 years of age in the preclinical dementia phase [43]. In accordance with that, in our study, subjects who reached a score lower than 95% of the maximum score had a higher BMI, as well as a higher percentage of body fat (Figure 1). Several hypotheses could explain this phenomenon; one of them could be due to lower production of subclinical thyroid hormones [44,45]. The underproduction of these hormones has been associated with a greater accumulation of fat and a slowdown in mental processing, among other disorders [46], which would explain the lower cognitive performance found in our patients with a higher BMI. Nevertheless, the most plausible explanation to that is the lack of physical exercise compared to those who have obtained better scores. This fact could be also explained due to the visible deficit in muscle mass in the group with lower cognitive score. In any case, physical activity has been shown to be a great tool in preserving cognitive status and preventing dementia [35,47].

Other studies have linked raised BMI levels, especially visceral fat, to a higher degree of oxidative stress in young adults [48]. Oxidative stress promotes a proinflammatory scenario that aims to minimize injuries, necrotic cells, and repair tissue damage [49]. Thus, aging promotes a low intensity inflammatory state, an excess of reactive oxygen species, and increases senescence and cellular dysfunction, which in turn, promotes development of neurodegenerative diseases [50,51]. The described process might be accelerated in people with obesity [52]. In agreement, our data revealed that the groups of people with lower cognitive performance (<95% of the maximum score) showed a BMI compatible with obesity (BMI > 30), while the group of people with higher scores in the mini-mental test had a normal BMI (see Figure 1a). 

According with previous studies, a higher percentage of visceral fat was detected in the groups with lower cognitive performance (see Figure 1b) [53]. A recent study has associated the visceral fat and brain cortical thickness in aged people, supporting the idea that visceral fat could be play a major role in the neurodegeneration induced by obesity [54]. The underly mechanisms to explain this relation could be that visceral fat accumulation is directly related to lipid homeostasis. This type of fat deposit generates greater oxidative stress, and the beginning of a pro-inflammatory process that promotes a lipid accumulation at the ectopic level [48,55,56]. This mechanism could explain the lower scores obtained in MMES associated with a higher percentage of visceral fat. On the other hand, obesity per se, regardless of associated vascular disease or insulin resistance, significantly increases the risk of dementia, specifically AD [57].

Here our data showed that as body fat increased, the percentage of muscle mass was decreased, in accordance with other works [40]. Previous clinical work associates the loss of muscle mass with a proinflammatory state, insulin resistance, and cognitive decline [58]. Aging and loss of muscle mass might be arisen from mitochondrial dysfunction [59]. During aging, there is a progressive myocytes reduction, which presents a heterogeneous distribution [60]; these changes lead to a loss of muscle mass and deterioration of muscle mobility that hinders physical exercise and promotes a more sedentary lifestyle. Moreover, the decrease in muscle mass could be due to incipient vascular damage caused by obesity that hind muscle respiration and accelerates aging and myocyte death. It is well established that an increment in visceral fat mass is a risk factor for ischemia or vascular dysfunction [61]. In this sense, aged muscle exhibits reduced aerobic function even when receiving similar levels of oxygen supply as young muscles [62].

When we determined the implication of metabolic parameters on cognition preservation in people over 60 years old, we observed that metabolic determinations including BMI, percentage of body fat, percentage of visceral fat, and percentage of muscle mass are good predictors of percentage of MMSE scores, supporting the role of the metabolic regulation in the preservation of cognitive abilities. Of note, previous studies in patients have also shown significant associations between obesity and brain alterations, detected by MRI [63], as well as between cortical and metabolic disturbances from an early age [64]. Furthermore, previous studies conducted in mouse models showed that worse metabolic conditions correlate with lower rates of central cell proliferation, increased neurogenesis rates, and progress in cognitive impairment [65,66]. It has been observed that during youth, the body system tries to compensate the generation of new neurons in young mice, but this ability is diminished in adults and the elderly, especially in the presence of a dementia such as AD [67]. Human studies have shown a similar trend and suggest that obesity and AD share the same neurodegenerative mechanisms, promoting in turn, an acceleration of the degenerative process at the central level [68].

Altogether, our limited studies suggest that obesity or excessive fat mass accelerates aging brain progression, including loss of mental agility, as well as loss of muscle mass, as reflected in the graphical abstract. Interestingly, metabolic parameters may partially predict many of these alterations; therefore, it is plausible that controlling metabolic parameters associated with obesity in old people could improve disease control and dementia prognosis.

### Study Limitations and Future Research

Since bioelectrical impedance analysis used in our study is not the gold standard, muscle mass or fat mass measures our results must be considered with caution. For future investigations, the gold standard measures, which provide us more sensitivity and specificity, will be considered, such as the proper assay to validate mass muscle, percentage of body fat, and visceral fat. An example of such an assay is the Dual energy X-ray. In addition, the present work has a limited number of participants. In this sense, it is necessary to carry out additional studies with a larger population in which the data obtained and presented here will be ratified. On the other hand, it would be interesting to evaluate how long-term interventions with diet control and increased physical activity in mentally active older people would affect the preservation of cognitive abilities.

## 5. Conclusions

We can conclude that a greater number of investigations are needed on dementia development in the mentally active elderly population, and a greater number of patients involved in this study is necessary to ratify the fat role as potential risk factors for dementia in this subpopulation. In this way, an increase in physical exercise and proper diet could be a suitable therapy to prevent the development of dementia. It will be useful to clarify the type of relationship between body composition and the rate of progression of mental aging to establish its role as a risk factor and methodology in the diagnosis and prognosis of dementia.

## Figures and Tables

**Figure 1 biomedicines-11-00138-f001:**
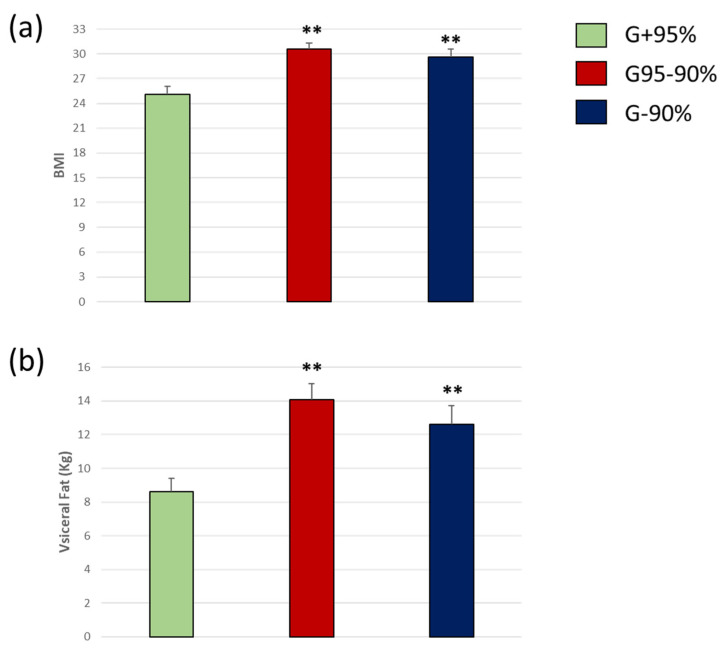
Body mass index (**a**) and visceral fat (**b**) in people older than 60 years with different mental agility conservation. ** *p* < 0.01 vs. G+95%. *p*-values for BMI: G + 95% vs. G95 − 90% *p* = 0.000, G + 95% vs. G − 90% *p* = 0.003 and G95 − 90% vs. G − 90% *p* = 0.433; Visceral Fat (kg): G + 95% vs. G95 − 90% *p* = 0.000, G + 95% vs. G − 90% *p* = 0.010 and G95 − 90% vs. G − 90% *p* = 0.332.

**Figure 2 biomedicines-11-00138-f002:**
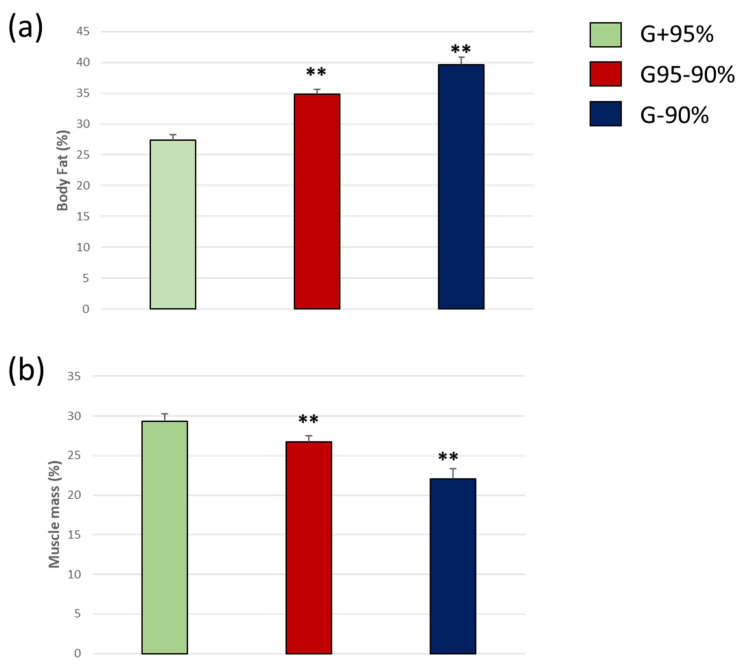
Body mass index (**a**) and visceral fat (**b**) in people older than 60 years with different mental agility conservation. ** *p* < 0.01 vs. G+95%. *p*-values for Body Fat (%): G + 95% vs. G95 − 90% *p* = 0.001, G + 95% vs. G − 90% *p* = 0.010 and G95 − 90% vs. G − 90% *p* = 0.258; Muscle Mass (%): G + 95% vs. G95 − 90% *p* = 0.038, G + 95% vs. G − 90% *p* = 0.000 and G95 − 90% vs. G − 90% *p* = 0.008.

**Figure 3 biomedicines-11-00138-f003:**
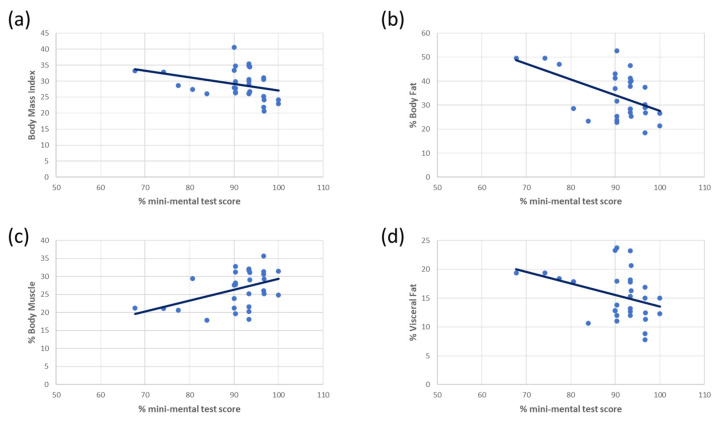
Metabolic parameters are good predictors of scores obtained in the MMSE.

**Table 1 biomedicines-11-00138-t001:** Sex and age distribution per groups.

Groups	Ages	Female (n)	Male (n)	Total (n)
G+95%	73.87 ± 9.249	10	10	19
G95 − 90%	73.58 ± 9.097	12	14	26
G − 90%	82.2 ± 2.38	8	6	15

*p*-values for Ages: G + 95% vs. G95 − 90% *p* = 0.942, G + 95% vs. G − 90% *p* = 0.062 and G95 − 90% vs. G − 90% *p* = 0.052.

**Table 2 biomedicines-11-00138-t002:** Body Height, Body Weight, and Basal Metabolic Rate in the different MMSE percentage score-groups.

Groups	Body Height	Body Weight	BMI	Basal Metabolic Rate
G + 95%	165.69 ± 9.36 cm	69.49 ± 15.39 Kg	25.08 ± 0.98	1454.38 ± 253.22 Kcal/day
G95 − 90%	166.53 ± 8.21 cm	84.35 ** ± 12.56 Kg	30.61 ** ± 0.72	1570.53 ± 223.23 Kcal/day
G − 90%	155.70 ** ± 7.78 cm	71.98 ± 8.83 Kg	29.64 ** ± 0.93	1389.6 * ± 164.99 Kcal/day

** *p* < 0.01 vs. rest of the groups, * *p* < 0.05 vs. G95 − 90%. *p*-values for Body Height: G + 95% vs. G95 − 90% *p* = 0.760, G + 95% vs. G − 90% *p* = 0.008 and G95 − 90% vs. G − 90% *p* = 0.002; *p*-values for Body weight: G + 95% vs. G95 − 90% *p* = 0.002, G + 95% vs. G − 90% *p* = 0.605 and G95 − 90% vs. G − 90% *p* = 0.002; *p*-values for BMI: G + 95% vs. G95 − 90% *p* = 0.000, G + 95% vs. G − 90% *p* = 0.003 and G95 − 90% vs. G − 90% *p* = 0.433; *p*-values for Basal Metabolic Rate: G + 95% vs. G95 − 90% *p* = 0.128, G + 95% vs. G − 90% *p* = 0.438 and G95 − 90% vs. G − 90% *p* = 0.011.

## Data Availability

Not applicable.

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
