# Peer review of "Poor Cognitive Agility Conservation in Obese Aging People"

_biomedicines, 2023, doi:10.3390/biomedicines11010138_

Round 1

Reviewer 1 Report

This study evaluated the metabolic composition and the cognitive abilities of a population 60-90 yrs, mentally healthy, with high academic degree. They found relationships between their cognitive abilities and their body composition: excess body fat was associated with poorer maintenance of cognition; higher percentage of muscle mass were associated with the best results in the cognitive tests.

I assume you are using the Mini Mental Test (MMT) first described by Folstein et al. I suggest that you add this reference, since it is one of the main instruments you use. Also start with the full name end then abbreviation (MMT).
Ref: Folstein MF. Mini Mental State. A practical method for grading cognitive status of patients for thr clinicians. J Psychiatr Res 1975; 21:189-198.

Scoring page 3. To me and other readers with experience of MMT and cognition/dementia I suggest that you also give the MMT scores between the different MMSE scores not only in %.

Group 1: scores higher than 95% of the maximum grade (G+95%) (n=16) 97 Group 2: scores between 95%-90% of the highest grade (G95-90%) (n= 10) 98 Group 3: scores less than 90% of the highest grade (G-90%) (n=34).

Table 1 I suggest that you add BMI for the 3 groups in one column!

It would have been interesting to examine if physical activity plays a role in this i.e if the difference is smaller in physical active obese elderly compared to nonphysical active obese; non-obese?

I suggest that you may add the FINGER-study from Finland to the references

The language needs to be checked and corrected; I can give some examples

Page 2; Line 51-56
Moreover, obese young adults have impaired the working memory to healthy weight controls [15]. ?

Line 61 The mini-mental test is simple – The mini-mental test is a simple
An a is missing

Line 69  “mechanisms that may underlying this obesity-cognitive”  - underly

Line 75 “ punctuation” – suggest performance

Line 80 resulted reliable predictors – in

Line 101 summone – what does it mean??

Line 103-105 Subsequently, the percentage of muscle mass, percentage of body fat, percentage of visceral fat and “metabolic consumption” were measured using a magnetic bioimpedance scale (OMRON Healthcare BF 511). Then, ???

Results:

Page 6, line 171  that increased body??? Is

Line 185 and body somatotype???

Author Response

We would like to thank you for your helpful comments on the manuscript. We are sincerely grateful to you for the time spent reading, commenting, and suggesting changes and modifications, which have greatly improved the quality of the work. We believe that the revisions made based on their reports have significantly improved the original manuscript.

Reviewer 1

This study evaluated the metabolic composition and the cognitive abilities of a population 60-90 yrs, mentally healthy, with high academic degree. They found relationships between their cognitive abilities and their body composition: excess body fat was associated with poorer maintenance of cognition; higher percentage of muscle mass were associated with the best results in the cognitive tests.

I assume you are using the Mini Mental Test (MMT) first described by Folstein et al. I suggest that you add this reference, since it is one of the main instruments you use. Also start with the full name end then abbreviation (MMT).

Ref: Folstein MF. Mini Mental State. A practical method for grading cognitive status of patients for thr clinicians. J Psychiatr Res 1975; 21:189-198.

Response: the version of MMT used in the methods has been included: “the participants were subdivided into three groups based on the results obtained by the NORMACODEM MMSE version [22], based in the original version described by Folstein et al. [23].”

Scoring page 3. To me and other readers with experience of MMT and cognition/dementia I suggest that you also give the MMT scores between the different MMSE scores not only in %.

Response: we have added an explication for these: NORMACORM MMSE version applies a score correction based on the years of academic studies and age (over or under 75 years), this causes the maximum score are slightly different in people over 75 years than younger population. In order to correct for these slight differences in scores for different ages, were have placed the scores as a reference of 100% of the maximum grade that the patients potentially can obtained.”

Group 1: scores higher than 95% of the maximum grade (G+95%) (n=16) 97 Group 2: scores between 95%-90% of the highest grade (G95-90%) (n= 10) 98 Group 3: scores less than 90% of the highest grade (G-90%) (n=34).

Table 1 I suggest that you add BMI for the 3 groups in one column!

Response: thanks for your comment. It was done

It would have been interesting to examine if physical activity plays a role in this i.e if the difference is smaller in physical active obese elderly compared to nonphysical active obese; non-obese?

Response: thanks for your comment. we did not in the questionnaire carried out previously to all the participants, we found no differences in physical activity between the different groups.

I suggest that you may add the FINGER-study from Finland to the references

Response: Thank you very much for this comment, it has been very enriching to learn about this project and the results it is offering. We have incorporated in the introduction a text where we include it. “In this sense, FINGER-study was one of the first researches to show that a multidomain lifestyle intervention is beneficial for prevention of cognitive decline [22]. Since 2009, when this project began, there have been refuted that the strategies based on change a mul-tidomain lifestyle intervention prevent cognitive decline [23,24].”

The language needs to be checked and corrected; I can give some examples

Page 2; Line 51-56

Moreover, obese young adults have impaired the working memory to healthy weight controls [15]. ?

Response: thanks for your comment. We Changed this sentence by: “Furthermore, it seems that some studies show that obese young adults have impaired the working memory”

Line 61 The mini-mental test is simple – The mini-mental test is a simple

An a is missing OK

Line 69  “mechanisms that may underlying this obesity-cognitive”  - underly OK

Line 75 “ punctuation” – suggest performance OK

Line 80 resulted reliable predictors – in OK

Line 101 summone – what does it mean?? OK

Line 103-105 Subsequently, the percentage of muscle mass, percentage of body fat, percentage of visceral fat and “metabolic consumption” were measured using a magnetic bioimpedance scale (OMRON Healthcare BF 511). Then, ??? OK

Results:

Page 6, line 171  that increased body??? Is OK

Line 185 and body somatotype??? OK

Response: Thank you very much for your comments, we have reviewed and modified all your assessments and others more.

We have reviewed and adapted it to the modifications suggested by the reviewers, in order to improve the quality of our paper and make it more attractive and complete for the reader. Hopefully this new version of the article will be now suitable for publication in Biomedicines

Author Response

We would like to thank you for your helpful comments on the manuscript. We are sincerely grateful to you for the time spent reading, commenting, and suggesting changes and modifications, which have greatly improved the quality of the work. We believe that the revisions made based on their reports have significantly improved the original manuscript.

Reviewer 2

General questions:

  • Is the manuscript clear, relevant for the field and presented in a well-structured manner?

Yes, the manuscript is relevant for the field, but could be presented in a clear manner.

Response: Thanks for the comment, we have restructured the discussion based on your recommendations.

·           Are the cited references mostly recent publications (within the last 5 years) and relevant? Does it include an excessive number of self-citations?

23 out of 52 references are older than 5 years; nevertheless, the references used are appropriate and relevant. I believe that other references could be included that are also relevant (this issue will be discussed later in the comments to the discussion).

Response: Thank you for your comment and for your contributions, we have added the citations proposed and others more

·           Is the manuscript scientifically sound and is the experimental design appropriate to test the hypothesis?

The aims of the manuscript are to correlate the performance in the MMSE with body composition in a sample of individuals older than 65 years old. The hypothesis is reasonable and interesting, but the design of the study presents several limitations. First, I believe that 60 individuals are not sufficient to make inferences to the population. With this study design, more individuals should be included. The authors do not mention if a sample size calculation was carried out.

Response: thanks for the comments, as we have included in the discussion, we know that the number of samples is limited. However, we have evaluated 100% of the people who met the above inclusion criteria: people over 60 years of age, mentally active and with higher education belonging to our university.

Second, the assessment of Body composition was made using the OMRON Healthcare BF 511 which is questionable method to assess body fat. Many of the works using this method are published in grey literature or in obscure publications; nevertheless, even those publications question the reliability and accuracy of this method and do not recommend it to be used in research setting.

Response: We agree that there are more precise methods to measure BMI than the OMRON Healthcare BF 511, however recent studies published in prestigious journals use this same methodology:

Madden KM, Feldman B, Arishenkoff S, Meneilly GS. A rapid point-of-care ultrasound marker for muscle mass and muscle strength in older adults. Age Ageing. 2021 Feb 26;50(2):505-510. doi: 10.1093/ageing/afaa163. PMID: 32909032; PMCID: PMC7936023

Cramer H, Hohmann C, Lauche R, Choi KA, Schneider N, Steckhan N, Rathjens F, Anheyer D, Paul A, von Scheidt C, Ostermann T, Schneider E, Koppold-Liebscher DA, Kessler CS, Dobos G, Michalsen A, Jeitler M. Effects of Fasting and Lifestyle Modification in Patients with Metabolic Syndrome: A Randomized Controlled Trial. J Clin Med. 2022 Aug 14;11(16):4751. doi: 10.3390/jcm11164751. PMID: 36012990; PMCID: PMC9410059.

Third, MMSE is a reliable tool to assess general cognition, but it is insufficient to discriminate between cognitive domains (memory or executive function for instance). A more detailed cognitive characterization should be performed.

As the reviewer 2 comment MMSE is a reliable tool to assess general cognition, We know that there are tests that allow a more exhaustive study of the different mental skills to be carried out, but these have not been carried out during the data collection and we do not have these data. We appreciate your comment and we will introduce new cognitive studies in future works.

·           Are the manuscript’s results reproducible based on the details given in the methods section?

The details given in the methods section are sufficient to reproduce the research that is presented, but no sufficient data on the sample characteristics is presented to ensure that the same characteristics are reproduced.

Response: Thank you very much for your comments.  We are aware of the sample size, but we have evaluated all the individuals of the subpopulation under study, as we described in the methodology and in the discussion.

·           Are the figures/tables/images/schemes appropriate? Do they properly show the data? Are they easy to interpret and understand? Is the data interpreted appropriately and consistently throughout the manuscript?

Figures are good and tables are also fine. I believe that some data that should be presented is not presented, and this is a strong limitation to the interpretation of the study. The statistics used is appropriate, but they use two strategies that make them present the “same” results two times (see comments to results).

Response: thanks for your comments.

·           Are the conclusions consistent with the evidence and arguments presented?

I believe that the conclusions are somehow abusive considering the results presented (se discussion comments).

Response: thanks for your comment. we have softened the conclusions by adding: our limited studies suggest…

·           Are the ethics statements and data availability statements adequate?

No data availability statement is presented. The ethics statements statement is provided and adequate.

Response: thanks for your comments

Article General concept comments:

In overall this is a work supported by an interesting hypothesis. The study design presents some flaws and limitations.

Response: thanks for your comments. We have tried to improve our study according to your comments.

Specific comments

The work needs some improvement in language editing and grammar. Introduction:

Throughout the text the authors mention the Minimental-test or the mini-mental test. The correct name is Mini Mental State Examination, commonly abbreviated to MMSE. Please correct.

Response: thanks for your comments. Minimental-test has been changed by MMSE

The authors present some demographic information that is valuable. They also present the association between obesity and chronic diseases, nevertheless the focus given to cancer seems misplaced. I think that would be more interesting to explore the associations of obesity comorbidities and cognitive decline (metabolic syndrome and cognition, insulin resistance and cognition, inflammation and cognition). When the authors present the inconsistent results regarding obesity and cognition, they do not point the limitations of the studies and do not propose a possible reason that may link to their goal. For instance, the way the cognitive assessment is made may be different from the strategy used in this work and could have prevented them to find results in line with the results expected.

Response: thanks for your comments. We are eliminated the cancer from introduction and we add the next text in the introduction:

“In this sense, insulin receptors are widely distributed throughout the central nervous system [9]. They are located in astrocytes, endothelial cells and neuronal synapses, they are very abundant in the hippocampus, cortex and cerebellum, protecting neurons from neurodegeneration and cell death, thus affecting learning and memory processes. Insulin resistance associated with obesity could alter the balance of all these processes through (for review [10]).

From another point of view, insulin resistance detected in obesity patients could be mediated by a decrease in the activity of the enzymes responsible for its degradation, as insulin degrading enzyme and neprylisin. These both enzymes are involved not only in insulin degradation but also in amyloid-β degradation in the central nervous system. am-yloid-β accumulation a one of first event in the AD development. In this sense, it has been postulated that an imbalance of substrates can affect the degradation rate of other sub-strates and possibly influence the pathogenesis of AD promoting an increase of amyloid-β accumulation and neuronal death [11].”

I would like that introduction could guide me to understand why the authors chose to use the strategy they used regarding the sample- and age-adjustment MMSE score.

Response: thanks for your comments. Previous studies indicate that the relationship between obesity-cognitive impairment is only evident in people over 65-70 years of age. That is why we have selected this age range. To try to clarify this point we have added the following text in the introduction:

“Following this idea, it has also been shown that every 1% rise in body mass index (BMI), at age of 70 years, could increase Alzheimer’s disease (AD) risk by 36% [5]. Therefore, the risk to develop cognitive impairment can increase by obesity in people over 65 years of age [6]. In that sense, hypercaloric diets have been related with insulin resistance, prediabetes and T2D.

I have serious concerns with the following sentence:

“Taken together, due to the unequivocal absence of evidence demonstrating an association between obesity and cognitive impairment function, a more in-depth study is needed to evaluate the mechanisms that may underlying this obesity-cognitive dysfunction axis. In this sense, the aim of this study was to…”

This study does not explore mechanisms and certainly is not an in-depth study.

Response: thanks for your comments. the mentioned text has been removed and replaced by: “Taken together, due to the presence of studies that relate cognitive impairment with body composition in mentally active older people are limited”

Material and Methods

The authors indicate the following:

“The scores ob-93 tained were expressed as a percentage, corrected by the maximum grade given to each

94 age group, which was placed as a reference of 100% of the grade.”

Interesting strategy but we need to know what the age groups.

Response: we have added an explication for these: NORMACORM MMSE version applies a score correction based on the years of academic studies and age (over or under 75 years), this causes the maximum score are slightly different in people over 75 years than younger population. In order to correct these slight differences in scores for different ages, were have placed the scores as a reference of 100% of the maximum grade that the patients potentially can obtained.”

The sample size is small for this kind of work, do the authors made any sample size calculation to estimate the sample size needed to have statistical power?

Response: Thank you very much for your comments, which are useful to us to improve our methodology in future works. We understand the limitations of our study and the limited size of the sample. Although we strongly believe in the relevance of the data. to try to address this topic we have added the following text in the second paragraph of the discussion: An important fact that we must consider when evaluating our results obtained is their origin and the sample number. We are aware sample number is limited by the number of people between 60 and 90 years of age, mentally active and without diagnosed cognitive pathologies enrolled in the permanent classroom for the elderly people at the University of Granada on the Ceuta Campus. Even though having a higher education level and have a mental activity are a preventive factor for suffering from dementia, they do not avoid its development [28]. That is why is relevant to study the other factors that may contribute to cognitive decline in people who remain mentally active.”

In section 2.3 the authors mention that they used magnetic bioimpedance, I believe they wanted to say bioelectric impedance. Also, the equipment OMRON Healthcare BF 511 is an electrical bioimpedance monitor. I never heard about magnetic bioimpedance.

The use of a scale such as OMRON Healthcare BF 511 is a concern for me regarding the reliability and accuracy of the results. As an example, Brtková, M., Bakalár, P., Matúš, I., Hančová, M., & Rimárová, K. (2014). Body composition of undergraduates – comparison of four different measurement methods. Physical Activity Review, (2), 38–44.

Response: Thank you very much for your comments. Section 2.3 has been changed and the citation has been added.

No mention to the cognitive assessment is made in the material and methods section. It is important to understand if the MMSE was applied by a trained interviewer.

Response: Thank for your comment. The cognitive assessment methodology has been added in the 2.2 section. “the participants were subdivided into three groups based on the results obtained by the NORMACODERM MMSE version [27], based in the original version described by Folstein et al. [28].”

Results:

The authors present the participants characteristics regarding body composition and basal metabolic rate, but the characterization regarding education and gender distribution are not presented and those are important characteristics. For instance, cognitive performance is associated with education and education is also associated with obesity. Differences observed could be due to differences in education.

Response: Thank for your comment. We have added a new text in the result and we added a new table with the ages and sexes per group. “As expected, all the participants had higher education level, all of them are studying the permanent elderly class at the University or were employees of the University. 50% of participants were female and 50% Male. The distribution of sexes and ages in each group can be seen in table 1. No differences were found in ages per group (p=0.91).”

Table 1. Sex and age distribution per groups

Groups

Ages

Female

Male

Total

G+95%

73.87 ± 9.249

10

8

18

G95-90%

73.58 ± 9.097

12

14

26

G-90%

82.2 ± 2.38

8

6

14

the authors present the result for the ANOVA (F) but should present also for the post hoc tests.

Response: Thank for your comment. We have added the post hoc test used in each statistical test.

It presented the results of the ANOVA comparing the body composition and metabolic rate by the groups of the MMSE, then it is also presented the correlation of the body composition and metabolic rate with the MMSE. This seems is only a different way to present the same result.

Response: Thank for your comment. We agree with your assessment. In the graphs we observe that there are intergroup differences in the measured anthropometric parameters. However, not always when there are differences, correlations are established. We believe it is important to demonstrate that an increase in body fat and loss of muscle mass are associated with worse cognitive preservation status. In the same way, we believe that we should highlight that these differences also show a progressive correlation, where %fat vs MMSE is negatively correlated and %musculature mass vs MMSE is positively correlated. In our opinion, these correlations strengthen the working hypothesis, making the relationship discussed stronger.

Discussion:

The authors do not discuss the results in light of previous findings. It would be expected to see the comparison of these results with results from other works. None of the results is compared with results from other works and the discussion is mainly composed by possible mechanisms associating cognition and body composition.

Response: We have added more similar studies and compared them with our data as you proposed. Many thanks for this recommendation which let us to improve our work.

A significant difference is found regarding height between groups and a previous report has already explored this (Pereira VH, Costa PS, Santos NC, Cunha PG, Correia-Neves M, Palha JA, Sousa N. Adult Body Height Is a Good Predictor of Different Dimensions of Cognitive Function in Aged Individuals: A Cross-Sectional Study. Front Aging Neurosci. 2016 Sep 16;8:217. doi: 10.3389/fnagi.2016.00217.) and put foward possible explanations.

Response: Thank you very much for this comment. We have incorporated this interesting cite in the discussion and we have added a discussion for height differences detected between the different groups.

 We have reviewed and adapted it to the modifications suggested by the reviewers, in order to improve the quality of our paper and make it more attractive and complete for the reader. Hopefully this new version of the article will be now suitable for publication in Biomedicines

Round 2

Reviewer 2 Report

See file

Author Response

New responses were added in red

As I have previously stated, many of the works using this device are published in obscure journals, obviously not all.

My concern is not with the prestigious of the publications but with the accuracy of the method. Indeed, those two publications are good references, despite the second one use it but does not assess its accuracy (the aim of the study is other) and in the first one they state the following:

“Limitations and future research

 The technique used in our study to measure LBM was BIA, which is not the gold standard measure of muscle mass (DXA scans). Future work on establishing PoCUS- based markers of muscle mass will need to be validated against more gold standard muscle measures, such as DXA. Such measures will be necessary to determine the sensitivity and specificity of these measures compared to current gold standard EWGSOP criteria.”

This kind of statement is important, and a similar statement should be added to this work. The authors must acknowledge that BIA (OMRON Healthcare BF 511) is not a gold standard, may not be accurate and results should be interpreted with caution.

Response: thank you for your comment. We are aware that the technique used to assess body composition is not the gold standard for this type of measurement. We have added a new paragraph where we address this point in results:

3.4. Study limitations and future research

From Bioelectrical impedance analysis used in our study is not the gold standard muscle mass or fat mass measures our results should be interpreted with caution. In the future investigations our data of mass muscle, percentage of body fat and percentage of visceral fat should be validated by the gold standard measures which provide us more sensitivity and specificity as Dual energy X‐ray absorptiometry. Also, the present work has a limited number of participants. In this sense, it is necessary to carry out additional studies with a larger population in which the data obtained and presented here will be ratified. On the other hand, it would be interesting to evaluate how long-term interventions with diet control and increased physical activity in mentally active older people would affect the preservation of cognitive abilities.

In section 2.3 the authors mention that they used magnetic bioimpedance, I believe they wanted to say bioelectric impedance. Also, the equipment OMRON Healthcare BF 511 is an electrical bioimpedance monitor. I never heard about magnetic bioimpedance.

This was not changed and is a mistake that must be corrected.

Response: Thanks for pointing this bug out and sorry we didn't fix it in the first review. “Magnetic bioimpedance” has been replaced by “bioelectrical impedance analysis” throughout the manuscript.

The use of a scale such as OMRON Healthcare BF 511 is a concern for me regarding the reliability and accuracy of the results. As an example, Brtková, M., Bakalár, P., Matúš, I., Hančová, M., & Rimárová, K. (2014). Body composition of undergraduates – comparison of four different measurement methods. Physical Activity Review, (2), 38– 44.

Response: Thank you very much for your comments. Section 2.3 has been changed and the citation has been added.

 Citations were added, but no other alterations were made to the section. Please correct magnetic bioimpedance.

Response: Thanks for pointing this bug out and sorry we didn't fix it in the first review. “Magnetic bioimpedance” has been replaced by “bioelectrical impedance analysis” throughout the manuscript.

No mention to the cognitive assessment is made in the material and methods section. It is important to understand if the MMSE was applied by a trained interviewer.

Response: Thank for your comment. The cognitive assessment methodology has been added in the 2.2 section. “the participants were subdivided into three groups based on the results obtained by the NORMACODERM MMSE version [27], based in the original version described by Folstein et al. [28].”

No mention to the qualification of the interviewer that applied the MMSE is made. It was a psychologist or at least a trained interviewer?

Response: Thanks for your comment. We have added a sentence where we explain who made the MMSE in line 111-112: This test was performed…. by a single nurse trained by a psychologist specializing in dementia.

the authors present the result for the ANOVA (F) but should present also for the post hoc tests.

Response: Thank for your comment. We have added the post hoc test used in each statistical test.

This was not what should be mentioned, I want to know the p-value of the post-hoc test. The p-value presented is for the ANOVA, I need the p-value for each post hoc comparison the p-value for G+95 vs G95-90, the p-value for the comparison G95- 90 vs G-90 and the p-value G+95 vs G-90.

Response: sorry for this and thanks for your clarification. We have added an additional text with p-values in the footer of tables and figures:

Table 1: G+95% vs G95-90% p=0.942, G+95% vs G-90% p=0.062 and G95-90% vs G-90% p=0.052;

Table 2: **p<0.01 vs G+95%.  P-values for Body Height: G+95% vs G95-90% p=0.760, G+95% vs G-90% p=0.008 and G95-90% vs G-90% p=0.002; Body weight: G+95% vs G95-90% p=0.002, G+95% vs G-90% p=0.605 and G95-90% vs G-90% p=0.002; BMI: G+95% vs G95-90% p=0.000, G+95% vs G-90% p=0.003 and G95-90% vs G-90% p=0.433; Basal Metabolic Rate: G+95% vs G95-90% p=0.128, G+95% vs G-90% p=0.438 and G95-90% vs G-90% p=0.011.

Figure 1: **p<0.01 vs G+95%.  P-values for BMI: G+95% vs G95-90% p=0.000, G+95% vs G-90% p=0.003 and G95-90% vs G-90% p=0.433; Visceral Fat (kg): G+95% vs G95-90% p=0.000, G+95% vs G-90% p=0.010 and G95-90% vs G-90% p=0.332.

Figure 2: **p<0.01 vs G+95%. P-values for Body Fat (%): G+95% vs G95-90% p=0.001, G+95% vs G-90% p=0.010 and G95-90% vs G-90% p=0.258; Muscle Mass (%): G+95% vs G95-90% p=0.038, G+95% vs G-90% p=0.000 and G95-90% vs G-90% p=0.008.

The authors do not discuss the results in light of previous findings. It would be expected to see the comparison of these results with results from other works. None of the results is compared with results from other works and the discussion is mainly composed by possible mechanisms associating cognition and body composition.

Response: We have added more similar studies and compared them with our data as you proposed. Many thanks for this recommendation which let us to improve our work.

 I believe that is not yet in accordance with the guidelines of the journal Please see the following:

“Discussion: Authors should discuss the results and how they can be interpreted in perspective of previous studies and of the working hypotheses. The findings and their implications should be discussed in the broadest context possible and limitations of the work highlighted. Future

research directions may also be mentioned. This section may be combined with Results.”

Response: We have added 2 new sections in which we explain the limitations and future research that we should carry out to validate the data obtained in this work: 3.4. Study limitations and future research.

3.4. Study limitations and future research

Since bioelectrical impedance analysis used in our study is not the gold standard muscle mass or fat mass measures our results must be considered with caution. For future investigations, the gold standard measures, which provide us more sensitivity and specificity, will be considered as the proper assay to validate mass muscle, percentage of body fat and visceral fat instead of Dual energy X‐ray absorptiometry. Also, the present work has a limited number of participants. In this sense, it is necessary to carry out additional studies with a larger population in which the data obtained and presented here will be ratified. On the other hand, it would be interesting to evaluate how long-term interventions with diet control and increased physical activity in mentally active older people would affect the preservation of cognitive abilities.

We have a added a conclusions section:

  1. Conclusions

We can conclude that a greater number of investigations are needed on dementia development in mentally active elderly, and a greater number of patients involved in this study is necessary to ratify the fat role as potential risk factors for dementia in this subpopulation. In this way, an increase in physical exercise and proper diet could be a suitable therapy to prevent the development of dementia. It will be useful to clarify the type of relationship between body composition and the rate of progression of mental aging to establish its role as a risk factor and methodology in the diagnosis and prognosis of dementia.